# Airway Bacterial Biodiversity in Exhaled Breath Condensates of Asthmatic Children—Does It Differ from the Healthy Ones?

**DOI:** 10.3390/jcm11226774

**Published:** 2022-11-16

**Authors:** Kamil Bar, Paulina Żebrowska, Łukasz Łaczmański, Barbara Sozańska

**Affiliations:** 11st Department and Clinic of Paediatrics, Allergology and Cardiology, Wroclaw Medical University, 50-367 Wroclaw, Poland; 2Laboratory of Genomics and Bioinformatics, Hirszfeld Institute of Immunology and Experimental Therapy, Polish Academy of Sciences, 53-114 Wroclaw, Poland

**Keywords:** asthma, microbiome, exhaled breath condensate

## Abstract

Asthma etiopathology is still not fully determined. One of its possible causes can be found in airway microbiome dysbiosis. The study’s purpose was to determine whether there are any significant differences in the bacterial microbiome diversity of lower airways microbiota of asthmatic children, since knowledge of this topic is very scarce. To the authors’ knowledge, this is the first research using exhaled breath condensates in children’s lower airways for bacterial assessment. Exhaled breath condensates (EBC) and oropharyngeal swabs were obtained from pediatric asthmatic patients and a healthy group (*n* = 38, 19 vs. 19). The microbial assessment was conducted through genetic material PCR amplification, followed by bacterial 16S rRNA amplicon sequencing. Collected data were analyzed, in terms of taxonomy and alpha and beta diversity between assessed groups. Swab samples are characterized by higher species richness compared to exhaled breath condensates (Shannon diversity index (mean 4.11 vs. 2.867, *p* = 9.108 × 10^−8^), observed features (mean 77.4 vs. 17.3, *p* = 5.572 × 10^−11^), and Faith’s phylogenetic diversity (mean 7.686 vs. 3.280 *p* = 1.296 × 10^−10^)). Asthmatic children had a higher abundance of bacterial species (Shannon diversity index, mean 3.029 vs. 2.642, *p* = 0.026) but more even distribution (Pielou’s evenness, mean 0.742 vs. 0.648, *p* = 0.002) in EBC than healthy ones; the same results were observed within pediatric patients born naturally within EBC samples. In children with a positive family history of allergic diseases, alpha diversity of lower airway material was increased (Shannon’s diversity index *p* = 0.026, Faith’s phylogenetic diversity *p* = 0.011, observed features *p* = 0.003). Class Gammaproteobacteria and Bacilli were less abundant among asthmatics in the exhaled breath samples. The most dominant bacteria on a phylum level in both sample types were Firmicutes, followed by Proteobacteria and Actinobacteriota. The obtained outcome of higher bacterial diversity of lower airways among asthmatic patients indicates a further need for future studies of microbiota connection with disease pathogenesis.

## 1. Introduction

Asthma is a heterogeneous disease of which etiopathology is not yet fully determined [1]. As defined by The Global Initiative for Asthma, the chronic airway inflammation causes variable expiratory airflow limitations that can provoke symptoms such as wheezes, shortness of breath, chest tightness, and cough [2]. It is the most frequent chronic lung disease among the pediatric population and its, and other allergic diseases, morbidity is still rising, especially in developing countries [3].^⁠^ One of the possible causes is to be found in the biodiversity hypothesis. The lack of frequent contact with various environmental microbes, typical for modern civilization, disturb enriching the human microbiome and, by that, a proper immunological response, thus promoting inflammation and allergic reactions [4].⁠ On the other hand, certain pathogens are known as asthma triggers, such as RSV and rhinovirus infection [5]. Latest asthma prevention interventions promote natural colonization and advice to reduce iatrogenic factors that may lead to early life dysbiosis, underlying the impact of microbiota in asthma pathogenesis [6].

For many years, lower airways were thought to be sterile because standard cultures did not show any significant growth of pathogens in healthy people [7]. The improvement of microorganism detection through culture-free identification methods, such as high-throughput sequencing, declined that thesis, showing that the bronchial microbiome exists. Studies on adults have proven that, in asthmatic patients, there are qualitative and quantitative differences in the bacterial composition of the lower airway, especially a higher abundance of Proteobacteria, with a lower abundance of Bacteroides and Firmicutes [8].

Exhaled breath condensate (EBC) is a new potential material for the detection of bacteria in lower airways in children. It is a non-invasive method in which a sample is obtained from tidal breathing through a cooled collection device, on which surface droplets of airway water condense. Its usefulness in asthma studies has been widely proven in search of inflammation/metabolomic changes, but as for microbial studies, no specific recommendations were proposed [9].

In terms of pediatric patients, the data about upper respiratory tract microbiota and its impact on asthma is a topic of many studies, whereas knowledge of lower airways microbiome is limited, due to the invasive form of specimen collection during bronchoscopy, which, in most cases, requires anesthesia (protected brush specimen/bronchoalveolar lavage), and the noninvasive one—induced sputum—is not sensitive enough⁠ [10]. There are only a few studies to date that investigated both the upper and lower respiratory tracts in children and, due to small groups of participants, there are no strong conclusions yet to be found about their mutual relevance in children’s asthma.

To further investigate the biodiversity of upper and lower airways among children with asthma, in our study, we collected exhaled breath condensates and oropharyngeal swabs for lower and upper airways bacterial abundance and composition comparison. We aimed to assess if there are any significant differences between asthmatic and non-asthmatic children and between upper and lower respiratory tracts, as well as if there are any additional factors connected with airway bacterial abundance in asthmatic and non-asthmatic pediatric patients.

## 2. Materials and Methods

### 2.1. Study Participants

Our research was planned as a case-control study. The study group comprised of patients randomly recruited in an Allergology Outpatient Clinic of 1st Department of Pediatrics, Allergology, and Cardiology at the WroclawMedical University, Poland. A total of 19 children aged 6–17 with asthma diagnosed by an allergologist were enrolled in a study group, whereas the control group consisted of 19 healthy children within the same age range (Table 1). The exclusion criteria for both groups were: any infection and antibiotic treatment 30 days prior to the enrollment and any serious chronic diseases (besides asthma, allergic rhinitis, atopic dermatitis, and allergic conjunctivitis in the study group). All patients’ asthma were assessed as well controlled by the time of the study, 68.4% (*n* = 13) were treated with asthma medications daily, of which, inhaled corticosteroids daily intake patients were 52.63% (*n* = 10). A total of 89.47% (*n* = 17) asthmatics were characterized as atopic asthma, with sIgE/SPT proven inhalant allergy. The Human Investigation Ethics Committees at the Wroclaw Medical University approved the study. Written informed consent was obtained from all patients prior to inclusion. All participants were asked to fulfill an authorial questionnaire, regarding their general social status and health.

### 2.2. Sample Collection

From all of the participants’, oropharyngeal swabs and exhaled breath condensates were obtained. Firstly, a child was asked for a quick mouthwash with a bottled mineral water to rinse additional bacterial flora from oral cavity epithelium, then, in a sitting position, a swab was taken from a posterior wall of pharynx with sterile, non-cotton swabs without transport medium. Secondly, a participant was asked for mouthwash and to gargle with 5% sodium bicarbonate solution, as similar ones have been recommended in the previous studies [11].⁠ Then, an exhaled breath condensate was acquired, according to European Respiratory Society (ERS)/American Thoracic Society (ATS) recommendations [9]. For our study, we used the Rtube collection system (Respiratory Research, Inc., Austin, TX, USA), which consists of the collection tube, the mouthpiece with a saliva trap and one way-valve, and a cooling sleeve with an insulating cover. A participant was asked to breathe tidally in a sitting position, with a nose clip on during the procedure, for 10 min through a mouthpiece, as such time is recommended by the producer for children’s sampling. The cooling sleeve, prior to the procedure, was cooled down to −30 °C. By the end of the collection, the tube containing EBC was immediately sealed from both sides with rubber caps included by the manufacturer and subsequently stored in the freezer (−20 °C) until analysis. Due to insufficient DNA material, not all of the samples were possible to analyze; from the asthmatics group, 19 EBC and 12 swabs were included, and from the control group, 14 EBC and 14 swabs were included.

### 2.3. Genetic Material Isolation and Amplification

#### 2.3.1. DNA Extraction

Bacterial DNA extraction was performed using the QIAamp PowerFecal Pro DNA Kit (QIAGEN, Hilden, Germany). For the exhaled breath condensate (EBC), standard protocol was modified as follows: the steps sample preparation, cell lysis, and inhibitor removal technology were omitted. The first step in extracting DNA from EBC samples was, therefore, adding 600 µL of high-concentration salt solution CD3 to an average of 700 µL of the sample. The manufacturer’s instructions were followed from that step (bind DNA, wash, elute). Elution was performed in a volume of 50 µL of C6 solution. DNA extraction of throat swabs was performed according to the manufacturer’s instructions, only no bead tubes were used, and instead of this step, the swab was vortexed in 800 µL of lysis buffer (CD1) and incubated at room temperature for 30 min.

#### 2.3.2. Preparation of DNA Libraries for Sequencing

Oropharyngeal swab DNA libraries from DNA samples extracted from throat swabs were prepared according to the manufacturer’s instructions [12], using the kit QIAseq 16S/ITS Region Panel (QIAGEN, Hilden, Germany) targeting 16S variable region V3V4. The concentration of the prepared libraries was measured using Quantus Fluorometer (Promega Corporation, Madison, WI, USA), and that was tested using TapeStation4200 (Agilent Technologies, Inc., Santa Clara, CA, USA), High Sensitivity D1000 ScreenTape (Agilent Technologies, Inc., Santa Clara, CA, USA). All normalized libraries were pooled in the same volume to generate an equivalent number of raw reads with each library. The final pool was diluted at a concentration of 10 pM and used for sequencing.

EBC Preparation of DNA libraries using the above protocol was not successful; therefore, an alternative approach was used as described below. During the following PCR amplification, KAPA HiFi HotStart ReadyMix PCR Kit (Roche, Basel, Switzerland) was used. As a first, the entire 16S rRNA gene was amplified using the following primers forward primer [5′-AGAGTTTGATCMTGGCTCA-3′] and reverse primer [5′-AAGGAGGTGWTCCARCC-3′], under the conditions: 95 °C for 3 min, followed by 35 cycles of 98 °C for 20 s, 65 °C for 15 s, 72 °C for 72 s, and 72 °C for 5 min, 12 °C hold. The template for this reaction was total DNA extracted from EBC samples in a volume of 10 µL. Following the manufacturer’s guideline of library preparation [12] during the second PCR reaction, 16S rRNA V3 and V4 regions were amplified using primers: 16S Amplicon PCR Forward Primer [5′-TCGTCGGCAGCGTCAGATGTTGTATAAGAGACAGCCTACGGGNGGCWGCAG-3’] and 16S Amplicon PCR Reverse Primer [5′-GTCTCGTGGGCTCGGAGATGTGTATAAGAGACAGGACTACHVGGGTATCTAATCC-3’], with the untreated products of the first PCR reaction as a template. In that step, Illumina adapter sequences were attached to amplicons (forward overhang 5′ TCGTCGGCAGCGTCAGATGTGTATAAGAGACAG-[locus-specific sequence]; reverse overhang 5′ GTCTCGTGGGCTCGGAGATGTGTATAAGAGACAG-(locus-specific sequence)). The conditions for the reaction were: 95 °C for 3 min, followed by 25 cycles of 95 °C for 30 s, 55 °C for 30 s, 72 °C for 30 s, and 72 °C for 5 min, 12 °C hold. The products of the second PCR reaction were purified using KAPA Pure Beads. The next step was to attach Nextera DNA CD Indexes (Illumina, Inc., San Diego, CA, USA), and NEBNext Q5 Hot Start HiFi PCR Master Mix (New England Biolabs, Inc., Ipswich, MA, USA) was used in this step, according to the standard protocol. The concentration of the purified amplicons with Illumina adapters and indices was measured using Quantus Fluorometer (Promega Corporation, Madison, WI, USA), and the quality of prepared libraries was tested using TapeStation4200 (Aligent Technologies, Inc., Santa Clara, CA, USA), High Sensitivity D1000 ScreenTape (Agilent Technologies, Inc., Santa Clara, CA, USA). Similarly, normalized libraries were pooled in the same volume. The final pool was diluted at a concentration of 6 pM and used for sequencing.

#### 2.3.3. 16S rRNA Gene Sequencing

Oropharyngeal swab sequencing was performed using Illumina MiSeq, v3 cartridge 600 cycles, flow cell v3, according to producer protocol.

EBC Sequencing was performed using Illumina MiSeq, v2 cartridge 500 cycles, flow cell v2 nano, according to producer’s protocol. PhiX input was 6 pM, 5%.

### 2.4. Data Analysis

Bioinformatics analysis of microbiota was performed using QIIME2 2021.8 [13] and additional plugins, including other methods of microbiome analysis. Demultiplexing was performed using a custom script that uses cutadapt [14] to cut V3V4 primer sequences. Evaluation of the quality of the reads was performed using the summary method from the demux plugin on the artifact constructed from previously demultiplexed data. Paired-end sequencing was performed, but due to insufficient read quality, which made it impossible to combine forward and reverse reads, it was decided to cut forward read to 70 nt and use trimmed single-end read for further analysis. Trimming, denoising, dereplication, and chimera filtering was performed with the dada2 plugin for single-end reads [15]. The multiple sequences alignment was performed using mafft [16], and then the phylogenetic tree was constructed using fasttree [17]. Before the construction of the phylogenetic tree, highly variable positions were masked. All these steps are implemented as a single routine align-to-tree-mafft-fasttree from q2-phylogeny plugin. The q2 diversity plugin was used to estimate alpha (Shannon’s diversity index, observed features, Faith’s phylogenetic diversity [18], and Pielou’s evenness [19]) and beta diversity (Jaccard distance, Bray-Curtis dissimilarity, weighted UniFrac [20], unweighted UniFrac [21], and performed principal coordinate analysis (PCoA). The samples were rarefied (subsampled without replacement) to 16,830 sequences per sample. The PERMANOVA was performed to test beta diversity group significance, Kruskal–Wallis for alpha diversity group significance, and Spearman’s rank correlation coefficient for continuous sample metadata columns, *p*-value < 0.05 was considered statistically significant. The naïve Bayesian classifier was used to assign taxonomy to ASVs. It was trained using the method fit-classifier-naive-Bayes from feature-classifier plugin [22,23] on fragments of the 16S rRNA gene sequences derived from the SILVA 138 SSURef NR99 [24,25] database, where the sequences were aligned to the primer of the V3V4 fragment and trimmed to a length of 70 nt, as in the case of the analyzed sequences. ANCOM [26], which is a tool that identifies differentially abundant features in a group of samples, was used as implemented in the q2-composition plugin with default parameters.

## 3. Results

Microbiota of the lower (EBC) and upper (swab) respiratory tract varied, as statistically significant differences in alpha and beta diversity can be found between swab (*n* = 26) and EBC samples (*n* = 33).

Higher richness of microbiota is found in swab samples, compared to EBC, as measured by the Shannon diversity index (*p* = 9.108 × 10^−8^), observed features (*p* = 5.572 × 10^−11^), Faith’s phylogenetic diversity (*p* = 1.296 × 10^−10^), swab vs. EBC, respectively, and Kruskal–Wallis test (Figure 1A,B,D, respectively). The distribution of taxa is not significantly different in swab and EBC samples, as measured by Pielou’s evenness (*p* = 0.109, Figure 1C). Ecological distances differ between these groups of samples (Figure 2).

Similarly, in the asthma group alone, swab (*n* = 12) vs. EBC (*n* = 19) higher richness was observed in swab samples in Shannon diversity index (*p* = 1.907 × 10^−4^, Appendix A) and observed features (*p* = 3.651 × 10^−6^, Appendix A), as well as higher phylogenetic diversity (Faith’s, *p* = 3.777^−6^, Appendix A); contrarily, the distribution measured by Pielou’s evenness did not differ significantly between these two samples type (*p* = 0.068, Appendix A).

The ecological distances differ between swab (*n* = 12) and EBC (*n* = 19) from asthmatic patients in Jaccard (*p* = 0.001), Bray-Curtis (*p* = 0.001), unweighted UniFrac (*p* = 0.001) and weighted UniFrac (*p* = 0.001) metrics, PERMANOVA test, and 999 permutations.

In terms of the alpha diversity of healthy controls alone, oropharyngeal swabs (*n* = 14) samples were found to have higher bacterial diversity (Shannon’s diversity index *p* = 1.648 × 10^−4^, observed features *p* = 6.607 × 10^−6^, Faith’s phylogenetic diversity, *p* = 1.927 × 10^−5^), compared to exhaled breath condensates (*n* = 14, Appendix A).

The ecological distances differ between swab (*n* = 14) and EBC (*n* = 14) in healthy control group in Jaccard (*p* = 0.001), Bray-Curtis (*p* = 0.001), unweighted UniFrac (*p* = 0.001) and weighted UniFrac (*p* = 0.001) metrics, PERMANOVA test, and 999 permutations.

### 3.1. Exhaled Breath Condensate Samples Analysis

Statistically significant differences in alpha diversity were found between the asthmatic and control EBC samples.

Higher richness of microbiota (Shannon diversity index, *p* = 0.026) and more even bacterial distribution (Pielou’s evenness, *p* = 0.002) were found in the EBC from asthmatic patients, compared to healthy controls (*n* = 19 vs. *n* = 14, Figure 3). No relevant changes were found in other alpha diversity metrics, observed features, *p* = 0.854, mean 17.4 (±4.8) vs. 17.4 (±4.4), Faith’s phylogenetic diversity, *p* = 0.362, 3.370 (±0.807) vs. 3.150 (±0.839), asthmatic (*n* = 19) vs. control (*n* = 14), respectively, and Kruskal–Wallis test.

Statistically significant differences in beta diversity can be found between asthmatic (*n* = 19) and control (*n* = 14) EBC samples. Ecological distances differ between these groups of samples in Jaccard (*p* = 0.012), Bray-Curtis (*p* = 0.038), weighted UniFrac (*p* = 0.043) metrics, PERMANOVA test, and 999 permutations (Figure 4). 

Some differences were also observed within the EBC sample type, due to factors such as smoking or exposure to cigarette smoke, route of delivery, family burden of allergic diseases, medications taken, and age. The older the probant, the less diverse microbiota of lower respiratory tract (alpha diversity in the Shannon metric, r = −0.357, *p* = 0.041), and with age, it tended to lead to a less even distribution of taxa (Pielou’s evenness, r = −0.378, *p* = 0.030), all EBC samples (*n* = 33), and Spearman’s rank correlation coefficient. In children born with a naturally lower airway, bacterial microbiota (*n* = 17) were found to be more diverse (Shannon’s diversity index *p* = 0.002, Faith’s phylogenetic diversity *p* = 0.003) and more evenly distributed (Pielou’s evenness, *p* = 0.003) than in children born through C-section (*n* = 16, Figure 5). They can be distinguished as separate groups in terms of the route of delivery (beta diversity, Jaccard, *p* = 0.019 and Unweighted UniFrac, *p* = 0.021; PERMANOVA, 999 permutations).

Factors found significant among all EBC samples (*n* = 33) were tested in the EBC from patients with asthma (*n* = 19) and healthy controls (*n* = 14) separately.

Microbiota of EBC from asthmatic patients with family burden of allergic diseases was more diverse than those patients without this risk factor (with family burden *n* = 14 vs. without family burden *n* = 4; Shannon’s diversity index *p* = 0.026, Faith’s phylogenetic diversity *p* = 0.011, observed features *p* = 0.003; Appendix A). Asthmatic patients with concurrent inhalant allergies (*n* = 17) tended to have slightly more uniformly distributed taxa (Pielou’s evenness *p* = 0.034), compared to asthmatic patients without allergies (*n* = 2; Appendix A). In beta diversity, the factors allowing for distinguishing internal groups within the asthmatics were cigarette passive smoke (exposed, *n* = 7; not exposed, *n* = 12, unweighted UniFrac, *p* = 0.015), family risk (yes, *n* = 14; no risk, *n* = 4, unweighted UniFrac, *p* = 0.043), medications taken (yes, *n* = 14; no, *n* = 5, weighted UniFrac, *p* = 0.017), and the place of living (village, *n* = 8; town, *n* = 11, weighted UniFrac, *p* = 0.007; unweighted UniFrac, *p* = 0.014).

Healthy controls born naturally (*n* = 4) had more diverse microbiota found in EBC, compared to those born by C-section (*n* = 10, Shannon’s diversity index *p* = 0.011; Faith’s phylogenetic diversity *p* = 0.034; Appendix A). No such relationship was found in the group of asthmatics; however, the inverted proportion was observed, as the group of asthmatic patients born naturally counted13 people, and those born by cesarean section was 6.

### 3.2. Oropharyngeal Swabs

No significant changes in alpha and beta diversity were found between swab samples from asthmatic patients (*n* = 14) and healthy control (*n* = 12).

Shannon diversity index, mean 4.232 (±0.844) vs. 4.021 (±0.749), *p* = 0.411, observed features, mean 78.667 (±20.624) vs. 76.786 (±16.494), *p* = 0.777, Faith’s phylogenetic diversity, mean 8.046 (±1.273) vs. 7.559 (±1.739), *p* = 0.918, Pielou’s evenness, mean 0.673 (±0.111) vs. 0.644 (±0.107), *p* = 0.440, asthmatic vs. healthy control, respectively, Kruskal–Wallis test. Jaccard *p* = 0.197, Bray-Curtis *p* = 0.733, unweighted UniFrac *p* = 0.18 and weighted UniFrac *p* = 0.913, PERMANOVA test, and 999 permutations.

### 3.3. Dominant Taxa

To assess the dominant class in each sample type, we searched for bacteria that were present in at least 50% of EBCs/swabs and had a mean relative abundance >1%.

Among exhaled breath condensates, on class level complied with above (in consecutive order): Bacilli, Gammaproteobacteria, Alphaproteobacteria, and Actinobacteria, all present in 100% of EBC samples (Table 2).

Within swab samples the most dominant class, in consecutive order, were: Bacilli, Gammaproteobacteria, Bacteroidia, Fusobacteriia, Actinobacteria, Negativicutes, Clostridia, Campylobacteria (Table 3).

### 3.4. Taxonomic Assignment

Both exhaled breath condensates and oropharyngeal swabs were assessed, ranging from phyla to genera. The most abundant phylum in both the upper (Figure 6B) and lower (Figure 6A) respiratory tracts was Firmicutes, followed by Proteobacteria and Actinobacteriota. There were differences in the compositions of microbiota in EBC samples. Using the ANCOM method [26], it was detected that at the class levels Gammaproteobacteria (Figure 7B) and Bacilli (Figure 7A) were the more abundant among healthy controls, compared to asthmatic patients (W = 1, both). No statistically relevant differences were found between asthmatics and the control group, both comparing swabs. As the ANCOM method assumes that the groups differ by less than 25% of features (bacteria), the EBC vs. swab was not performed.

## 4. Discussion

No statistically significant differences were found between asthmatics and the control group in oropharyngeal swabs, in terms of bacterial alpha and beta diversity, also the taxa of significantly different abundance could not be observed. The microbiota of the lower (tested in exhaled breath condensates) and upper (tested in swabs) respiratory tracts varied. In our study, we found that asthmatic EBC samples were characterized by higher richness (Shannon’s diversity index) and more even distribution (Pielou’s evenness), in terms of alpha diversity. Beta diversity also showed relevant differences in the bacterial composition of this type of sample (Jaccard, Bray-Curtis, and weighted UniFrac), compared to healthy controls. Additionally, negative correlations of age with richness (Shannon index, *p* = 0.041) and evenness (Pielou’s evenness, *p* = 0.030) were observed; however, the correlations were low. More diverse bacterial microbiota, in terms of richness (Shannon index, Faith’s phylogenetic diversity) and evenness, were found in EBC samples of children born naturally. In terms of beta diversity, separate groups could also be observed (Jaccard and unweighted UniFrac). However, after the division into groups, among children from the control group, the above-mentioned relationships remained, but they could not be observed in the group of children with asthma. Notably, the asthmatics group was dominated by naturally born children, while the inverse proportion in the control group was observed as children born via cesarean section predominated in that group. The asthmatics with a family burden of allergic diseases in this study were discovered to have increased microbial richness, but this result should be treated with caution, as the number of people with asthma without a burden was very low and incomparable with the group of people with such a burden. Such disproportion can be explained by the fact of a high heritability of allergic diseases, ranging up to 95%, when assessing asthma [27]. Similarly, it appears that people with asthma and comorbid allergies had a more uniformly distributed microbiota, but also the size of the group was too small to draw such a conclusion. Factors affecting significant differences in beta diversity between EBC samples from asthmatic patients were exposure to cigarette smoke, family burden, inhalant corticosteroid intake, and place of living. The most abundant phyla in EBC samples were Firmicutes, Proteobacteria, and Actinobacteriota, and in swabs, Firmicutes, Proteobacteria, and Bacteroidetes. Among exhaled breath samples representing lower airways microbiota, the classes Gammaproteobacteria and Bacilli were found to be more abundant within the control group, compared to asthmatics. Asthma, and its connection with human microbiome dysbiosis, is still a topic of many researchers. To our present knowledge, the microbiota of the lower respiratory tract exists and differs from that of the upper respiratory tract.

The Perez-Losada et al. study assessed the nasopharyngeal microbiome of both children and adults asthmatics in a group of 40 patients, concluding that *Moraxella*, *Haemophilus*, *Staphylococcus*, *Streptococcus*, and *Fusobacterium* are the core genera of their upper respiratory tract [28].⁠ In his another study on children exclusively, five phyla were dominant in nasal swabs: Firmicutes, Proteobacteria, Actinobacteria, Bacteroidetes, and Fusobacteria, in descending order, which also comprises our EBC findings [29]. Additionally, in this particular research, including asthmatics only, varieties in genera composition among groups divided by certain phenotypes were observed, mainly in *Moraxella*, *Corynebacterium*, *Prevotella*, *Dolosigranulum*, and *Staphylococcus* abundance [29]. Depner et al.’s study, conducted in children with asthma, revealed lower alpha and beta diversity of nasal bacterial microbiota, whereas no difference was found within throat swabs, compared to healthy control [30]. Oropharyngeal swabs analyzed by Boutin et al., obtained from children with cystic fibrosis and asthmatics, revealed lower alpha diversity in the CF group, compared to asthmatics, but no difference was found comparing children with asthma to the healthy control [31]. In our study, no statistical relevance in the biodiversity of the upper respiratory tract has been observed within the compared groups.

Interestingly, studies on adults with the use of induced sputum were comparable with our findings. Marri et al.’s study showed that Firmicutes, Proteobacteria, and Actinobacteria accounted for more than 90% of the found sequences [32], which was similar to our taxonomical outcomes, despite other sample type sources. The same bacterial composition, with the addition of Bacteroidetes, accounted for 98% in the sputum samples of Pang et al.’s study; moreover, lower alpha diversity was observed within non-eosinophilic asthmatics [33].⁠ The same sample type in the Simpson et al.’s study revealed certain microbial dysbiosis, regarding the asthma phenotype. *Haemophilus influenzae* was found to be more abundant among neutrophilic asthmatics, whereas *Tropheryma whipplei*, a member of Actinobacteria phylum, was among the eosinophilic [34].

Research including the lower airway microbiome in asthmatic children is very scarce. The study by Hilty et al. was the first based on endobronchial brushings of adults and BAL samples in pediatric patients [35]. The adult groups (*n* = 24) comprised asthmatic, COPD patients, and healthy controls, whereas, among children, 13 were characterized as difficult asthma (defined as the need for at least 3 times a week rescue medications, besides high doses of inhaled and/or oral steroids), and 7 non-asthmatics underwent bronchoscopy for other indications. All of the participants had their nose and oropharyngeal swabs obtained, as well. Bronchial brushings were collected from the left upper lobe (LUL, *n* = 23) and right lower lobe (RLL, *n* = 14); not all of the patients tolerated right lobe samplings. The nasal microbiota clustered differently from other sample sites within all studied groups. Oropharyngeal microbiota of healthy controls clustered with their LUL samples and oropharyngeal swabs of asthmatics. A higher abundance of the Proteobacteria phylum and lower of Bacteroidetes were found in both the adult and pediatric asthmatics, as well as COPD adults in lower airways samples, compared to the controls. On a genus level, *Haemophilus* spp. were more abundant in asthmatics, regardless of age and COPD patients, additionally in the pediatric group higher richness of *Staphylococcus* spp. and lower of *Prevotella* spp. were found in asthmatics, compared to the controls. However, a small group of pediatric patients and concomitant diseases of “non-asthmatic” controls could have affected the outcomes. In another study by Goldman et al., bronchoalveolar lavage (BAL) of asthmatics was mainly enriched by the *Streptococcus* and *Prevotella* genera, members of Firmicutes and Bacteroidetes phyla, respectively [36].⁠ Cui et al. assessed bronchoalveolar lavage fluid’s bacterial richness in children with pneumonia. Patients with high total IgE levels (>60 UI/mL), including asthmatics and/or people suffering from allergic rhinitis, presented a higher abundance of *Bacteroides* and lower of *Streptococcus*, *Lactobacillus*, and *Anoxybacillus* [37]. The study’s main limitation was antibiotic therapy prior, or ongoing, to the sample collection, which probably altered microbial homeostasis. In preschool children, asthma diagnosis remains challenging, since there is no objective test to perform as disease confirmation [2]. Researchers assess children with recurrent or persistent wheezing as an asthma equivalent. In Robinsons et al.’s study of episodic viral and multi-trigger wheezers underwent bronchoscopy [38]. In bronchoalveolar lavage samples (group of 14 children), two distinct groups were assessed with *Moraxella* predominance in one, with the mean relative abundance of 47.5% and the “mixed” group with more diverse bacterial contribution of other genera, whereas Moraxella’s 1.2% was average. In episodic viral wheezers, the *Moraxella* higher abundance and greater neutrophilic BAL count was observed. Another recent study conducted by Wu et al. was a 2-year follow-up of children with persistent wheezing [39]. In BAL samples, the lower abundance of *Fusobacterium* and *Moraxella* with higher of *Elizabethkingia* and *Rothia* was observed, compared to the healthy control; the last two genera increased substantially in children with wheezing recurrence as the follow-up was performed [39]. Bisgaard et al. observed that infants’ hypopharynx colonized with *Moraxella catarrhalis*, *Streptococcus pneumoniae*, and *Haemophilus influenzae* at one month of age were more likely to be diagnosed with asthma at 5 years [40]. Although, in our study, asthmatic patients were characterized by a lower abundance of Gammaprotobacteria (including *Moraxella* and *Haemophilus* species) and Bacilli (including *Streptococcus* species), we cannot compare our findings with the ones on a phylum level.

We did not find any strong data supporting our finding that inhalant allergies affect bacterial diversity that can comply with our outcome of a more even distribution of bacteria in asthmatic patients EBC. 

Inhaled corticosteroids (ICS) in asthma treatment and their role in microbiome diversity remains uncertain. Huang et al.’s study showed no significant relevance in sputum alpha diversity between asthmatics with and without ICS on the bacterial microbiome [41]. Denner et al. using endobronchial brushing (EB) in asthmatic adults have found a significant decrease in both alpha and beta bacterial diversity among patients using oral and inhaled corticosteroids, compared to milder asthma types with ICS treatment only [42].

EBC samples tend to be more diverse, in the sense of the bacterial microbiome, rather than BAL. In Zhang et al.’s study, patients suffering from severe asthma with the need for using more than 2000 µg/d of inhaled beclomethasone presented a higher abundance of Firmicutes phyla, predominantly *Streptococcus* genera, and a minor increase in Proteobacteria, comparing to asthmatics with better disease control [43]. ICS treatment was also found to be less effective in patients with lower airways enriched with asthma-associated *Haemophilus* [44]. In the latter study, it was shown that fluticason treatment through inhalation in non-responsive patients caused a greater deviation in the bacterial flora of induced sputum and oral cavity samples [45]. In our study, the EBC of asthmatics prescribed with daily medication intake, including ICS, differed in beta diversity (weighted_unifrac), compared to asthmatics without regular basis treatment.

A cesarean section is proven to be an independent factor increasing the risk of asthma development [46,47], additionally making children prone to respiratory infections and obesity in the future [48]. Our findings indicate more diverse lower airway bacterial microbiota of children born naturally, which may support the earlier studies on microbiome diversity reduction as a risk factor for asthma development. Interestingly, in our study, children with asthma were delivered mostly by vaginal birth and healthy controls through C-section, but due to the relatively low participants group, that outcome cannot be taken as a negation of the above. 

Rural areas residents are not only characterized by lower asthma prevalence [49,50], but allergic diseases in general [51]. In Depner et al.’s study, asthmatic children’s nasal microbiota richness was inversely connected with *Moraxella OTU 1462*; however, no such observation regarded the ones living on farms [29]. Higher microbial richness with lower abundance of Streptococcae family in the indoor dust microbiota of farmhouses was found to have an impact on the prevalence of asthma among children [52]. In our EBC samples of the asthmatic group, two separate groups could be distinguished in beta diversity, taking the rural areas of living into account.

Passive smoking is also a factor that can alter the microbiome of the respiratory tract. In Bugova et al.’s study, children exposed to second-hand smoking were more frequently colonized by potentially pathogenic bacteria than healthy peers, when isolated from middle nasal meatus and nasopharynx with the use of standard cultivation methods [53]. In children with allergic rhinitis, exposed to passive smoking, bacterial microbiota were characterized by a lower Simpson index, compared to non-exposed patients; the two groups were also statistically different, when comparing them by beta diversity [54]. There are also some data regarding third-hand smoke, which is exposure to toxic chemicals accumulating in an indoor environment. Neonates admitted to intensive care unit gut microbiota showed a decrease in alpha diversity and a lower number of OTUs among children affected by smoke in their households [55]. Third-hand smoke can also alter the bacterial composition of an indoor environment, increasing its alpha diversity, as well as skin surfaces, such as outer ear [56]. In our study, in exhaled breath samples of asthmatic patients, passive smoking exposure was found to be a discriminant factor, in terms of beta diversity.

The study limitations were associated with the lack of standardization, in terms of the microbiological assessment of exhaled breath condensates [8].⁠ The sample itself remains challenging; usually, the material’s collected volume ranges between 0.5–1.5 mL, from which 99.99% is water [57]. Very little genetic material within each condensate puts an effort into its isolation and amplification. No salivary contamination has been assessed in the EBC samples, as reducing the amount of collected material caused a risk of decreasing the bacterial DNA within the condensate. Despite the small study group, compared to induced sputum research, the number of participants was relatively high, when compared to studies regarding BAL samples. Furthermore, the induced sputum taken from children is characterized by a lack of reliability [10]. By the time of DNA isolation, the previous recommendation regarding exhaled breath condensates failed to provide a sufficient amount of material, thereby a new method was implemented in our study, which significantly improves the methodology of the microbiome assessment in EBC, in our opinion.

## 5. Conclusions

This novel approach to noninvasive lower airways microbiome sample collection among children highlights the need for further DNA/RNA assessment in exhaled breath condensates, which is currently one of the promising noninvasive methods for a better understanding of airways microbiome and asthma pathology. To the authors’ knowledge, this is the first study assessing lower respiratory microbiome in children with the use of exhaled breath condensates. In our research, a higher bacterial diversity in EBC from lower airways was found in asthmatic patients. Furthermore, having a family member suffering from other allergic diseases has also been found to have an impact on bacterial variety. Future studies on larger groups may determine the specific type of bacteria that can be applied as a biological factor to prevent asthma.

## Figures and Tables

**Figure 1 jcm-11-06774-f001:**
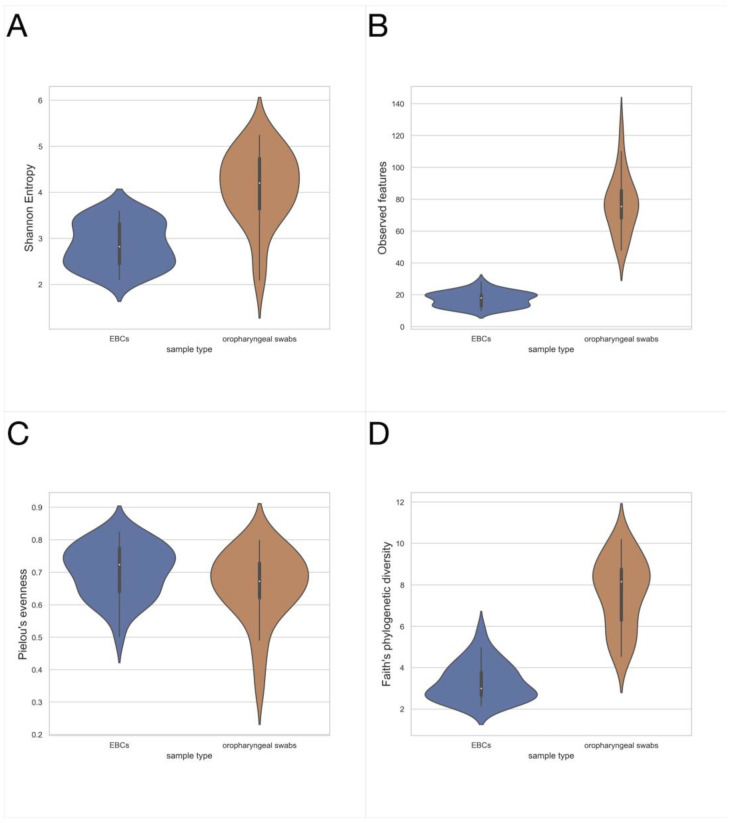
Alpha diversity metrics for sample type, asthmatic patients, and healthy controls taken together. (**A**) Shannon diversity index, mean 4.112 (±0.794) vs. 2.867 (±0.481), *p* = 9.108 × 10^−8^; (**B**) Observed Features, mean 77.4 (±18.1) vs. 17.3 (±4.4), *p* = 5.572 × 10^−11^; (**C**) Pielou’s evenness, mean 0.657 (±0.109) vs. mean 0.703 (±0.083), *p* = 0.109; (**D**) Faith’s Phylogenetic Diversity, mean 7.686 (±1.672) vs. 3.280 (±0.829), *p* = 1.296 × 10^−10^ swab (*n* = 26) vs. EBC (*n* = 33), respectively, Kruskal–Wallis test.

**Figure 2 jcm-11-06774-f002:**
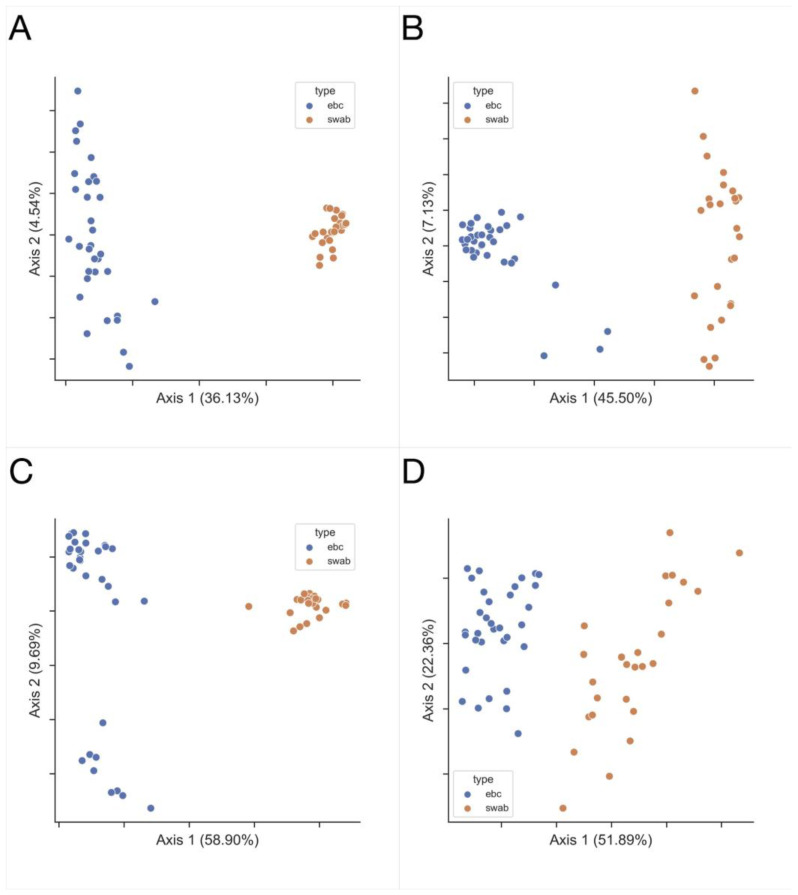
The microbiota of lower airways (EBC samples in blue, *n* = 33) differs from the microbiota of upper airways (swab samples, in orange, *n* = 26) asthma and control group taken together, as principal coordinate analysis (PCoA) plots capture the beta diversity for (**A**) Jaccard, (**B**) Bray-Curtis, (**C**) Unweighted UniFrac, and (**D**) Weighted UniFrac metrics, in all cases *p* = 0.001, PERMANOVA test, 999 permutations.

**Figure 3 jcm-11-06774-f003:**
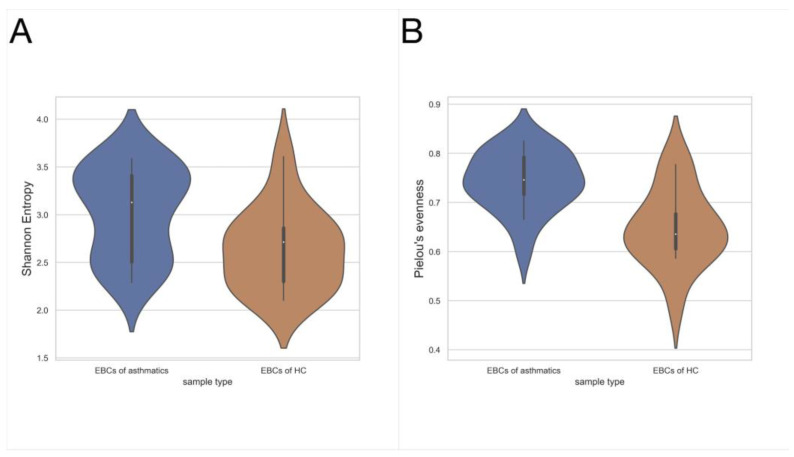
Alpha diversity metrics for EBC, asthmatic patients vs. healthy controls. (**A**) Shannon diversity index, mean 3.029 (±0.462) vs. 2.642 (±0.424), *p* = 0.026; (**B**) Pielou’s evenness, mean 0.742 (±0.060) vs. mean 0.648 (±0.078), *p* = 0.002; asthmatics (*n* = 19) vs. control (*n* = 14), respectively, Kruskal–Wallis test.

**Figure 4 jcm-11-06774-f004:**
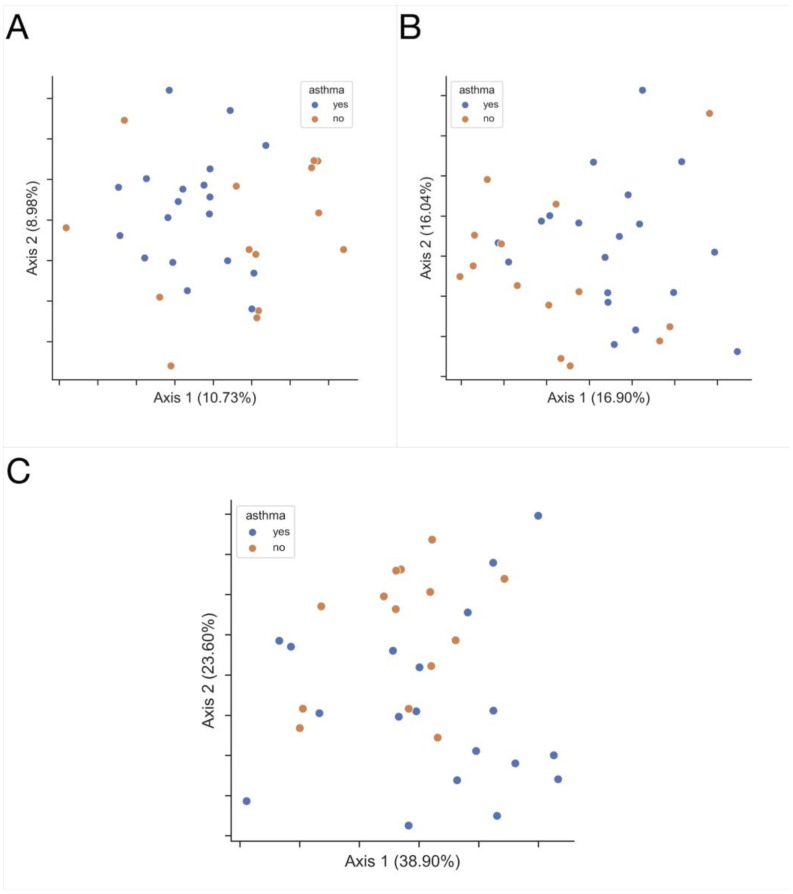
The microbiota of lower airways differs between asthmatics (blue) and HC (orange). Principal coordinate analysis (PCoA) plots capturing the beta diversity among EBC samples, for (**A**) Jaccard, *p*= 0.018; (**B**) Bray-Curtis, *p* = 0.039 and (**C**) Weighted UniFrac, *p* = 0.035.

**Figure 5 jcm-11-06774-f005:**
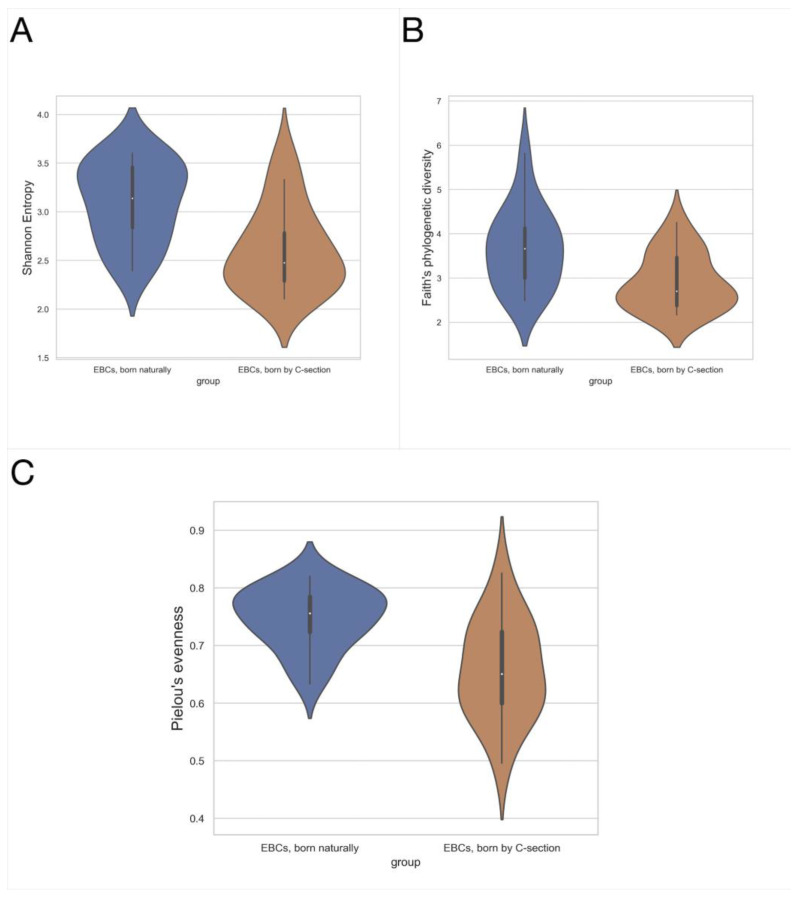
Alpha diversity metrics for EBC, children born naturally (*n* = 17) vs. born by C-section (*n* = 16). (**A**) Shannon diversity index, mean 3.1 (±0.409) vs. 2.615 (±0.434), *p* = 0.002; (**B**) Faith’s Phylogenetic Diversity, mean 3.675 (±0.907) vs. 2.91 (±0.637), *p* = 0.003; (**C**) Pielou’s evenness, mean 0.748 (±0.053) vs. 0.657 (±0.085), *p* = 0.003; Kruskal–Wallis test.

**Figure 6 jcm-11-06774-f006:**
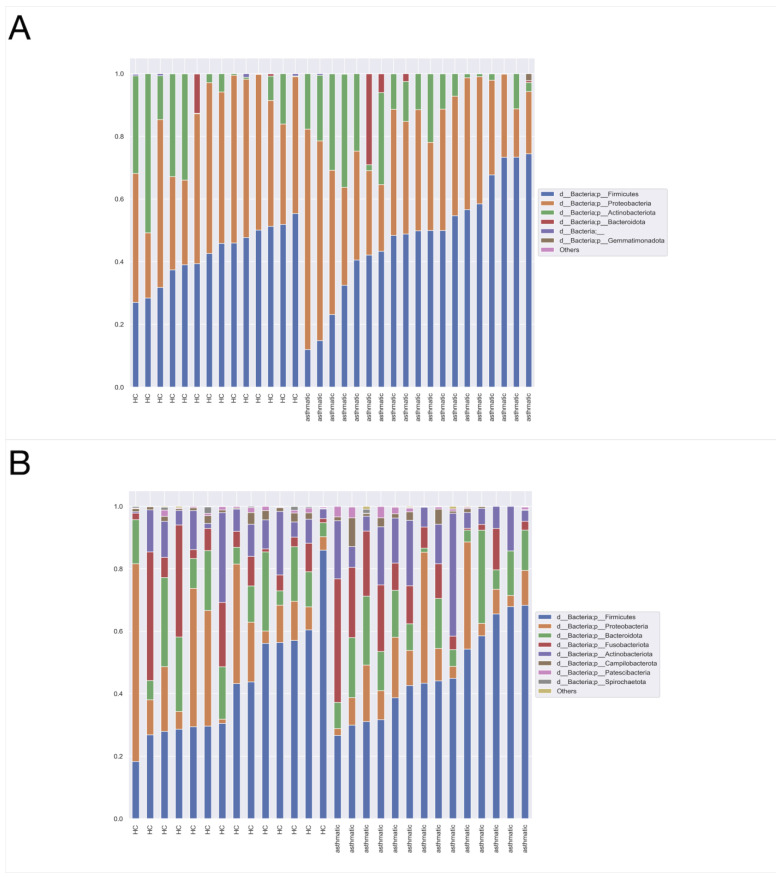
The relative abundance at the phylum level in lower airways—EBC samples (**A**); in upper airways—swab samples (**B**). d_—domain, p_—phylum.

**Figure 7 jcm-11-06774-f007:**
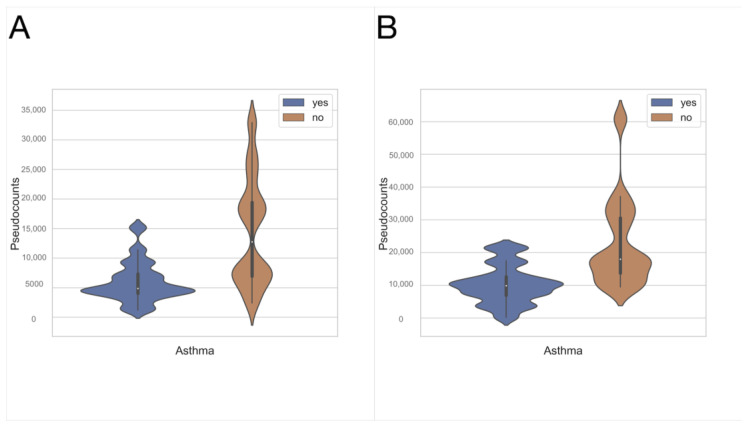
Relative abundance of differentially abundant taxa indicated by the ANCOM method, among EBC samples of asthmatics and healthy controls. The bacteria of class Bacilli (W = 1) (**A**) and Gammaproteobacteria (W = 1) (**B**) were more abundant in the group of asthmatics than in healthy controls.

**Table 1 jcm-11-06774-t001:** Study group characteristics.

	Asthmatic Group *n* = 19	Healthy Control *n* = 19
Swab collection	12 (63.16%)	14 (73.68%)
EBC ^†^ collection	19 (100%)	14 (73.68%)
Sex (women/men)	6 (31.58%)/13 (68.42%)	12 (63.16%)/7 (36.84%)
Mean age (years)	10.3	12
Place of living (town/village)	11 (57.89%)/8 (42.11%)	10 (52.63%)/9 (47.37%)
Overweight (>85 centile of BMI)	3 (15.79%)	3 (15.79%)
Underweight (<3 centile of BMI)	1 (5.26%)	1 (5.26%)
Asthma medications daily intake/Inhaled corticosteroids (ICS) treatment on a daily basis	14 (73.68%)/10 (52.63%)	none
Natural birth/cesarean section (cc)	13 (68.42%)/6 (31.58%)	7 (36.84%)/12 (63.16%)
Inhalant allergies	17 (89.47%)	none
Family history of allergic diseases	14 (73.68%)	none
Active smokers within household	7 (36.84%)	4 (21.05%)

^†^ EBC–exhaled breath condensate

**Table 2 jcm-11-06774-t002:** Dominant classes among EBC samples, in consecutive order.

Taxon	Sample Presence [%]	Mean Relative Abundance [%]
d__Bacteria;p__Firmicutes;c__Bacilli ^†,‡,§^	100.00	35.44
d__Bacteria;p__Proteobacteria;c__Gammaproteobacteria	100.00	20.92
d__Bacteria;p__Proteobacteria;c__Alphaproteobacteria	100.00	17.86
d__Bacteria;p__Actinobacteriota;c__Actinobacteria	100.00	13.72

^†^ d_—Domain, ^‡^ p_—Phylum, ^§^ c_—Class.

**Table 3 jcm-11-06774-t003:** Dominant classes among swab samples, in consecutive order.

Taxon	Sample Presence [%]	Mean Relative Abundance [%]
d__Bacteria;p__Firmicutes;c__Bacilli ^†,‡,§^	100.00	33.63
d__Bacteria;p__Proteobacteria;c__Gammaproteobacteria	100.00	16.63
d__Bacteria;p__Bacteroidota;c__Bacteroidia	100.00	13.35
d__Bacteria;p__Fusobacteriota;c__Fusobacteriia	96.43	11.71
d__Bacteria;p__Actinobacteriota;c__Actinobacteria	100.00	10.62
d__Bacteria;p__Firmicutes;c__Negativicutes	96.43	7.54
d__Bacteria;p__Firmicutes;c__Clostridia	92.86	3.71
d__Bacteria;p__Campilobacterota;c__Campylobacteria	92.86	1.83

^†^ d_—Domain, ^‡^ p_—Phylum, ^§^ c_—Class.

## Data Availability

The data presented in this study are available on request from the corresponding author.

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
