# Peer review of "Airway Bacterial Biodiversity in Exhaled Breath Condensates of Asthmatic Children—Does It Differ from the Healthy Ones?"

_jcm, 2022, doi:10.3390/jcm11226774_

Round 1
Reviewer 1 Report
I believe the Authors aimed to address a very important question and explained their research properly.
Some minor suggestions below:
- Abstract: it is a bit hard to follow, I would suggest to add some results numbers and a closing sentence with the main conclusion and potential implications of the results.
- Discussion: The discussion is very much focused on the results of the microbiome dysbiosis also in comparison with previous literature, which is an important point of discussion and it is very well written.
However, not too much discussion is spent for exhaled breath techniques, EBC is one, but there are also other non invasive ways to collect exhaled breath (such with the eNOSE or GC-MS analysis) and especially their potential future implications in asthma phenotyping. Are there so far any other studies that evaluate microbiome and breathomics in asthma in children?
To conclude, I think the topic and the study is valuable and could be an interesting starting point for future studies.
Author Response
Response to Reviewer 1 Comments
Point 1. Abstract: it is a bit hard to follow, I would suggest to add some results numbers and a closing sentence with the main conclusion and potential implications of the results.
Author response : Thank you for pointing it out. We have changed the abstract according to the mentioned suggestions, however it surpasses now recommended lenght by the journal (up to 200 words, currently 289)
The revised text reads as follow on lines 11-32 :
Abstract: Asthma etiopathology is still not fully determined. One of its possible causes can be found in airway microbiome dysbiosis. The study’s purpose was to determine whether there are any significant differences in bacterial microbiome diversity of lower airways microbiota of asthmatic children, since knowledge of this topic is very scarce. To the authors’ knowledge, this is the first research using exhaled breath condensates in children’s lower airways bacterial assessment. Exhaled breath condensates (EBC) and oropharyngeal swabs were obtained from pediatric asthmatic patients and healthy group (n=38, 19 vs. 19). The microbial assessment was conducted through genetic material PCR amplification, followed by bacterial 16S rRNA amplicon sequencing. Collected data were analyzed in terms of taxonomy, alpha and beta-diversity between assessed groups. Swab samples are characterized by higher species richness compared to exhaled breath condensates (Shannon diversity index (mean 4,11 vs 2,867, p=9.108x10-8), Observed Features (mean 77,4 vs 17,3, p=5.572x10-11) and Faith’s Phylogenetic Diversity (mean 7,686 vs 3,280 p=1.296x10-10). Asthmatic children had a higher abundance of bacterial species (Shannon diversity index, mean 3.029 vs. 2.642, p=0.026) but more even distribution (Pielou’s evenness, mean 0.742 vs. 0.648, p=0.002) in EBC than healthy ones; the same results were observed within pediatric patients born naturally within EBC samples. In children with a positive family history of allergic diseases, alpha-diversity of lower airway material was increased (Shannon’s diversity index p=0.026, Faith’s phylogenetic diversity p=0.011, Observed Features p=0.003). Class Gammaproteobacteria and Bacilli were less abundant among asthmatics in exhaled breath samples. The most dominant bacteria on a phylum level in both sample types were Firmicutes, followed by Proteobacteria and Actinobacteriota. Obtained outcome of higher bacterial diversity of lower airways among asthmatic patients indicates a further need for future studies of microbiota connection with disease pathogenesis.
Point 2. Discussion: The discussion is very much focused on the results of the microbiome dysbiosis also in comparison with previous literature, which is an important point of discussion and it is very well written. However, not too much discussion is spent for exhaled breath techniques, EBC is one, but there are also other non invasive ways to collect exhaled breath (such with the eNOSE or GC-MS analysis) and especially their potential future implications in asthma phenotyping. Are there so far any other studies that evaluate microbiome and breathomics in asthma in children?
Author response : Unfortunatelly, we have not find any researches regarding using other non-invasive methods to collect exhaled breath in microbiome studies. It is however a highly interesting topic to investigate in the future.
Reviewer 2 Report
This is a highly original research paper on bacteriological biodiversity in exhaled breath condensate, conducted on pediatric bronchial asthma patients. The study was conducted in Poland. The bronchial asthma patients included were 19 children to adolescents aged 6-17 years. Nineteen normal health adjusted for age and place of residence are compared as a control group.
Samples are oral swabs and exhaled breath condensate. For the study of bacteriological diversity, the authors and their colleagues extracted bacterial DNA and based on DNA libraries and 16S ribosomal RNA gene sequences Identification is under consideration.
The results obtained are presented in graphs separately for the upper and lower respiratory tracts and analyzed using the alpha diversity index. The results showed no significant differences between asthmatic and normal subjects in oropharyngeal swabs. However, there were suggestive results for the lower and upper respiratory tracts.
However, as the authors state in their limitation, there seems to be a limitation in concluding the bacteriological biodiversity in bronchial asthma patients from the results of this small sample.
The reviewers commend this study for its highly original content as a study design, despite its sample size limitations. We would then like to ask the following questions.
Major1
In the introduction, please first clarify the authors' definition of bronchial asthma. Can childhood asthma and adolescent asthma be considered the same concept of disease?
Major 2
Asthma has allergic and non-allergic mechanisms. The subject case is not described in this regard; it would be desirable to set up a unified sample of special allergens for IgE values, but has this been considered? What about infections, e.g., history of RS virus?
Major 3
In bronchial asthma, is the bronchus of concern the upper or lower respiratory tract? Which do you think it is? You have described an adult case, but I would recommend citing it in more detail.
Major 4
The authors also discuss this as a limitation, but how many cases do you think are needed to make this study more evidence-based? How many cases are considered based on the study, for example, in similar virological review papers? Please let us know if you have your own opinion.
We have made four comments, but the reviewers find this study, which focuses on the bacterial flora of the lower respiratory tract, to be novel. The reviewers will consider it a novel paper that could serve as a model for additional research in the future. Please respond to the reviewers' comments and incorporate their input into your paper.
Best regards,
Dr. Reviewer
Author Response
Response to Reviewer 2 Comments
Major1
In the introduction, please first clarify the authors' definition of bronchial asthma. Can childhood asthma and adolescent asthma be considered the same concept of disease?
Author response : Thank you for pointing that out. We have included definition of asthma in the introduction, according to GINA. We agree that asthma phenotypes depend on age in children. Our study group did not include children below 5 years of age with childhood asthma. We clasified participants as asthmatic based on the diagnosis made by allergologist (doctor’s diagnosed asthma)
The added text reads as follow on lines 37-39:
As defined by The Global Initiative for Asthma, the chronic airway inflammation causes variable expiratory airflow limitation that can provoke symptoms such as wheezes, shortness of breath, chest tightness and cough[37].
Major 2
Asthma has allergic and non-allergic mechanisms. The subject case is not described in this regard; it would be desirable to set up a unified sample of special allergens for IgE values, but has this been considered? What about infections, e.g., history of RS virus?
Author response : Thank you for pointing that out. We did include questions about participants allergies in the questionnaire and added information about atopic asthmatics.
The added text reads as follow on lines 89-90:
68,34% (n=13) asthmatics were characterized as atopic asthma, with sIgE/SPT proven inhalant allergy.
We have not conducted laboratory test of specific IgEs during the research.
The exclusion criteria of including participants were any infections in last 4 weeks. Given the fact that assessed children were at least 6 years old, it was hard to gather objective information about RSV infection (recall bias), moreover it would be difficult to distinguish such cases based on IgG presence , as they are present in the most of the population at that age.
Major 3
In bronchial asthma, is the bronchus of concern the upper or lower respiratory tract? Which do you think it is? You have described an adult case, but I would recommend citing it in more detail.
Author response : Thank you for pointing it out. Regardless of the disease, bronchi remains anatomically and functionally a part of lower airways. Asthmatic symptoms are connected with constriction of bornchiolis since they are not supported with cartilage, but the chronic inflammation, which is the main reason of undergoing symptoms, affect all of the bronchi/bronchioli linings. As for the Hilty et al. study, we have cited it more thouroghly as requested.
The revised text reads as follow on lines 437-453 :
The study by Hilty et al was the first based on endobronchial brushings of adults and BAL samples on pediatric patients[34]. Adult groups (n=24) comprised asthmatic, COPD patients and healthy controls, whereas among children 13 were characterized as difficult asthma (defined as the need for at least 3 times a week rescue medications besides high doses of inhaled and/or oral steroids) and 7 non-asthmatics underwent bronchoscopy for other indications. All of the participants have their nose and oropharyngeal swabs obtained as well. Bronchial brushings were collected from left upper lobe (LUL, n=23) and right lower lobe (RLL, n=14); not all of the patients tolerated right lobe samplings. The nasal microbiota clustered differently from other sample sites within all studied groups. Oropharyngeal microbiota of healthy controls clustered with their LUL samples and oropharyngeal swabs of asthmatics. A higher abundance of Proteobacteria phylum and lower of Bacteroidetes were found in both adult and pediatric asthmatics as well as COPD adults in lower airways samples, compared to controls. On a genus level Haemophilus spp. were more abundant in asthmatics regardless of age and COPD patients, additionally in the pediatric group higher richness of Staphylococcus spp. and lower of Prevotella spp. were found in asthmatics compared to controls. However small group of pediatric patients and concomitant diseases of “non-asthmatic” controls could have affected the outcomes.
We have removed the following lines, of which the number of pediatric patients was misspelled, for which we apologise :
A higher abundance of Proteobacteria was found in children’s samples, additionally in genera level Haemophilus spp. and Staphylococcus spp. were found more abundant in both pediatric and adult asthmatics. Nevertheless, a small group of pediatric patients (8 children) with concomitant diseases and no information about previous antibiotic therapy, that could affect the airway microbiome, made that outcome questionable.
Major 4
The authors also discuss this as a limitation, but how many cases do you think are needed to make this study more evidence-based? How many cases are considered based on the study, for example, in similar virological review papers? Please let us know if you have your own opinion.
Authors response : Thank you for pointing that out. The main limitation considering the larger amount of patients is the cost of PCR/NGS analysis. We have found research on induced sputum and bacterial assessment of pediatric asthmatics aged 6-15, with total participants of 95[1]. Researches with use of BAL samples comprised of 20-54 children [2-5]. We have found one research on exhaled breath samples and virome on 67 children with diagnosed asthma aged 5-12 years [6]. Regarding EBC solely there is a research of mycobiome assessment on 67 adults, of which 47 were asthmatics [7].
[1] Kim YH, Jang H, Kim SY, Jung JH, Kim GE, Park MR, Hong JY, Kim MN, Kim EG, Kim MJ, Kim KW, Sohn MH. Gram-negative microbiota is related to acute exacerbation in children with asthma. Clin Transl Allergy. 2021 Oct 12;11(8):e12069. doi: 10.1002/clt2.12069. PMID: 34667591; PMCID: PMC8507365.
[2] Hilty, M.; Burke, C.; Pedro, H.; Cardenas, P.; Bush, A.; Bossley, C.; Davies, J.; Ervine, A.; Poulter, L.; Pachter, L.; Moffatt, M. F.; Cookson, W. O. C. Disordered Microbial Communities in Asthmatic Airways. 2010, 5 (1). https://doi.org/10.1371/journal.pone.0008578.
[3] An, S.; Warris, A.; Turner, S. Microbiome Characteristics of Induced Sputum Compared to Bronchial Fluid and Upper Airway Samples. 2018, 2 (August 2017), 1–8. https://doi.org/10.1002/ppul.24037.
[4] Goldman, D. L.; Chen, Z.; Shankar, V.; Tyberg, M.; Vicencio, A.; Burk, R. Lower Airway Microbiota and Mycobiota in Children with Severe Asthma. J. Allergy Clin. Immunol. 2018, 141 (2), 808-811.e7. https://doi.org/10.1016/j.jaci.2017.09.018.
[5] Kloepfer, K. M.; Deschamp, A. R.; Ross, S. E.; Peterson-Carmichael, S. L.; Hemmerich, C. M.; Rusch, D. B.; Davis, S. D. In Children, the Microbiota of the Nasopharynx and Bronchoalveolar Lavage Fluid Are Both Similar and Different. Pediatr. Pulmonol. 2018, 53 (4), 475–482. https://doi.org/10.1002/ppul.23953.
[6] Tovey ER, Stelzer-Braid S, Toelle BG, Oliver BG, Reddel HK, Willenborg CM, Belessis Y, Garden FL, Jaffe A, Strachan R, Eyles D, Rawlinson WD, Marks GB. Rhinoviruses significantly affect day-to-day respiratory symptoms of children with asthma. J Allergy Clin Immunol. 2015 Mar;135(3):663-9.e12. doi: 10.1016/j.jaci.2014.10.020. Epub 2014 Dec 2. PMID: 25476729; PMCID: PMC7173323.
[7] Carpagnano, G.E.; Malerba, M.; Lacedonia, D.; Susca, A.; Logrieco, A.; Carone, M.; Cotugno, G.; Palmiotti, G.A.; Foschino-Barbaro, M.P. Analysis of the Fungal Microbiome in Exhaled Breath Condensate of Patients with Asthma. Allergy Asthma Proc. 2016, 37, e41–e46
Round 2
Reviewer 2 Report
I would like to thank you for responding to all four comments provided by the reviewers.The definition of asthma was also cited from the GINA definition. The discussion of asthma for ages 5 and under is also acceptable. Discussion of allergic vs. non-allergic asthma is also fine. I also understood the limitation of the study. I hope that the comments of the other reviewers will help to make this paper a better revision.
Best regards,
Dr. Reviewer